# A Unified View of cGANs with and without Classifiers

**Si-An Chen**
National Taiwan University
d09922007@csie.ntu.edu.tw

**Chun-Liang Li**
Google Cloud AI
chunliang@google.com

**Hsuan-Tien Lin**
National Taiwan University
htlin@csie.ntu.edu.tw

## Abstract

Conditional Generative Adversarial Networks (cGANs) are implicit generative models which allow to sample from class-conditional distributions. Existing cGANs are based on a wide range of different discriminator designs and training objectives. One popular design in earlier works is to include a classifier during training with the assumption that good classifiers can help eliminate samples generated with wrong classes. Nevertheless, including classifiers in cGANs often comes with a side effect of only generating easy-to-classify samples. Recently, some representative cGANs avoid the shortcoming and reach state-of-the-art performance without having classifiers. Somehow it remains unanswered whether the classifiers can be resurrected to design better cGANs. In this work, we demonstrate that classifiers can be properly leveraged to improve cGANs. We start by using the decomposition of the joint probability distribution to connect the goals of cGANs and classification as a unified framework. The framework, along with a classic energy model to parameterize distributions, justifies the use of classifiers for cGANs in a principled manner. It explains several popular cGAN variants, such as ACGAN, ProjGAN, and ContraGAN, as special cases with different levels of approximations, which provides a unified view and brings new insights to understanding cGANs. Experimental results demonstrate that the design inspired by the proposed framework outperforms state-of-the-art cGANs on multiple benchmark datasets, especially on the most challenging ImageNet. The code is available at https://github.com/sian-chen/PyTorch-ECGAN.

## 1 Introduction

Generative Adversarial Networks [GANs; 10] is a family of generative models that are trained from the duel of a generator and a discriminator. The generator aims to generate data from a target distribution, where the fidelity of the generated data is "screened" by the discriminator. Recent studies on the objectives [2, 37, 29, 25, 36, 26, 38], backbone architectures [41, 50], and regularization techniques [13, 35, 51] for GANs have achieved impressive progress on image generation, making GANs the state-of-the-art approach to generate high fidelity and diverse images [3]. Conditional GANs (cGANs) extend GANs to generate data from class-conditional distributions [33, 39, 34, 16]. The capability of conditional generation extends the application horizon of GANs to conditional image generation based on labels [39] or texts [43], speech enhancement [32], and image style transformation [18, 53].

One representative cGAN is Auxiliary Classifier GAN [ACGAN; 39], which decomposes the conditional discriminator to a classifier and an unconditional discriminator. The generator of ACGAN is expected to generate images that convince the unconditional discriminator while being classified to

35th Conference on Neural Information Processing Systems (NeurIPS 2021).

the right class. The classifier plays a pivotal role in laying down the law of conditional generation for ACGAN, making it the very first cGAN that can learn to generate 1000 classes of ImageNet images [6]. That is, ACGAN used to be a leading cGAN design. While the classifier in ACGAN indeed improves the quality of conditional generation, deeper studies revealed that the classifier biases the generator to generate easier-to-classify images [45], which in term decreases the capability to match the target distribution.

Unlike ACGAN, most state-of-the-art cGANs are designed without a classifier. One representative cGAN without a classifier is Projection GAN [ProjGAN; 34], which learns an embedding for each class to form a projection-based conditional discriminator. ProjGAN not only generates higher-quality images than ACGAN, but also accurately generates images in target classes without relying on an explicit classifier. In fact, it was found that ProjGAN usually cannot be further improved by adding a classification loss [34]. The finding, along with the success of ProjGAN and other cGANs without classifiers [15, 4], seem to suggest that including a classifier is not helpful for improving cGANs.

In this work, we challenge the belief that classifiers are not helpful for cGANs, with the conjecture that leveraging the classifiers appropriately can benefit conditional generation. We propose a framework that pins down the roles of the classifier and the conditional discriminator by first decomposing the joint target distribution with Bayes rule. We then model the conditional discriminator as an energy function, which is an unnormalized log probability. Under the energy function, we derive the corresponding optimization term for the classifier and the conditional discriminator with the help of Fenchel duality to form the unified framework. The framework reveals that a jointly generative model can be trained via two routes, from the aspect of the classifier and the conditional discriminator, respectively. We name our framework **E**nergy-based **C**onditional **G**enerative **A**dversarial **N**etworks (ECGAN), which not only justifies the use of classifiers for cGANs in a principled manner, but also explains several popular cGAN variants, such as ACGAN [39], ProjGAN [34], and ContraGAN [16] as special cases with different approximations. After properly combining the objectives from the two routes of the framework, we empirically find that ECGAN outperforms other cGAN variants across different backbone architectures on benchmark datasets, including the most challenging ImageNet.

We summarize the contributions of this paper as:

- We justify the principled use of classifiers for cGANs by decomposing the joint distribution.
- We propose a cGAN framework, Energy-based Conditional Generative Adversarial Networks (ECGAN), which explains several popular cGAN variants in a unified view.
- We experimentally demonstrate that ECGAN consistently outperforms other state-of-the-art cGANs across different backbone architectures on benchmark datasets.

The paper is organized as follows. Section 2 derives the unified framework that establishes the role of the classifiers for cGANs. The framework is used to explain ACGAN [39], ProjGAN [34], and ContraGAN [16] in Section 3. Then, we demonstrate the effectiveness of our framework by experiments in Section 4. We discuss related work in Section 5 before concluding in Section 6.

## 2   Method

Given a $K$-class dataset $(x, y) \sim p_d$, where $y \in \{1 \ldots K\}$ is the class of $x$ and $p_d$ is the underlying data distribution. Our goal is to train a generator $G$ to generate a sample $G(z, y)$ following $p_d(x|y)$, where $z$ is sampled from a known distribution such as $\mathcal{N}(0, 1)$. To solved the problem, a typical cGAN framework can be formulated by extending an unconditional GAN as:

$$\max_D \min_G \sum_y \mathop{\mathbb{E}}_{p_d(x|y)} D(x, y) - \mathop{\mathbb{E}}_{p(z)} D(G(z, y), y) \tag{1}$$

where $G$ is the generator and $D$ is a discriminator that outputs higher values for real data. The choice of $D$ leads to different types of GANs [10, 2, 29, 8].

At first glance, there is no classifier in Eq. (1). However, because of the success of leveraging label information via classification, it is hypothesized that a better classifier can improve conditional generation [39]. Motivated by this, in this section, we show how we bridge classifiers to cGANs by Bayes rule and Fenchel duality.

## 2.1 Bridge Classifiers to Discriminators with Joint Distribution

A classifier, when viewed from a probabilistic perspective, is a function that approximates $p_d(y|x)$, the probability that $x$ belongs to class $y$. On the other hand, a conditional discriminator, telling whether $x$ is real data in class $y$, can be viewed as a function approximate $p_d(x|y)$. To connect $p_d(y|x)$ and $p_d(x|y)$, an important observation is through the joint probability:

$$\log p(x, y) = \log p(x|y) + \log p(y) \tag{2}$$
$$= \log p(y|x) + \log p(x). \tag{3}$$

The observation illustrates that we can approximate $\log p(x, y)$ in two directions: one containing $p(x|y)$ for conditional discriminators and one containing $p(y|x)$ for classifiers. The finding reveals that by sharing the parameterization, updating the parameters in one direction may optimize the other implicitly. Therefore, we link the classifier to the conditional discriminator by training both objectives jointly.

## 2.2 Learning Joint Distribution via Optimizing Conditional Discriminators

Since $p(y)$ is usually known a priori (e.g., uniform) or able to easily estimated (e.g., empirical counting), we focus on learning $p(x|y)$ in Eq.(2). Specifically, since $\log p(x, y) \in \mathbb{R}$, we parameterize it via $f_\theta(x)$, such as a neural network with $K$ real value outputs, where $\exp(f_\theta(x)[y]) \propto p(x, y)$. Similar parameterization is also used in exponential family [48] and energy based model [23]. Therefore, the log-likelihood $\log p(x|y)$ can be modeled as:

$$\log p_\theta(x|y) = \log \left( \frac{\exp (f_\theta(x)[y])}{Z_y(\theta)} \right) = f_\theta(x)[y] - \log Z_y(\theta), \tag{4}$$

where $Z_y(\theta) = \int_{x'} \exp (f_\theta(x')[y]) \, dx'$.

Optimizing Eq. (4) is challenging because of the intractable partition function $Z_y(\theta)$. Here we introduce the Fenchel duality [48] of the partition function $Z_y(\theta)$:

$$\log Z_y(\theta) = \max_{q_y} \left[ \mathbb{E}_{q_y(x)} [f_\theta(x)[y]] + H(q_y) \right]$$

where $q_y$ is a distribution of $x$ conditioned on $y$ and $H(q_y) = - \mathbb{E}_{x f \sim q_y(x)} [\log q_y(x)]$ is the entropy of $q_y$. The derivation is provided in Appendix A. By the Fenchel duality, we obtain our maximum likelihood estimation in Eq. (4) as:

$$\max_\theta \left[ \mathbb{E}_{p_d(x,y)} [f_\theta(x)[y]] - \max_{q_y} \left[ \mathbb{E}_{q_y(x)} [f_\theta(x)[y]] + H(q_y) \right] \right]. \tag{5}$$

To approximate the solution of $q_y$, in additional to density models, we can train an auxiliary generator $q_\phi$ as in cGANs to estimate $\mathbb{E}_{q_y(x)}$ via sampling. That is, we can sample $x$ from $q_\phi$ by $x = q_\phi(z, y)$, where $z \sim \mathcal{N}(0, 1)$. The objective (5) then becomes:

$$\max_\theta \min_\phi \sum_y \mathbb{E}_{p_d(x|y)} [f_\theta(x)[y]] - \mathbb{E}_{p(z)} [f_\theta(q_\phi(z, y))[y]] - H(q_\phi(\cdot, y)), \tag{6}$$

which is almost in the form of Eq (1) except the entropy $H(q_\phi(\cdot, y))$. We leave the discussion about the entropy estimation in Section 2.4. Currently, the loss function to optimize the objective without the entropy can be formulated as:

$$\mathcal{L}_{d_1}(x, z, y; \theta) = -f_\theta(x)[y] + f_\theta(q_\phi(z))[y]$$
$$\mathcal{L}_{g_1}(z, y; \phi) = -f_\theta(q_\phi(z, y))[y]$$

## 2.3 Learning Joint Distributions via Optimizing Unconditional Discriminators & Classifiers

Following Eq. (3), we can approximate $\log p(x, y)$ by approximating $\log p(y|x)$ and $\log p(x)$. With our energy function $f_\theta$, $p_\theta(y|x)$ can be formulated as:

$$p_\theta(y|x) = \frac{p_\theta(x, y)}{p_\theta(x)} = \frac{\exp(f_\theta(x)[y])}{\sum_{y'} \exp(f_\theta(x)[y'])},$$

which is equivalent to the $y$'th output of $\texttt{SOFTMAX}(f_\theta(x))$. Therefore, we can maximize the log-likelihood of $p_\theta(y|x)$ by consider $f_\theta$ as a softmax classifier minimizing the cross-entropy loss:

$$\mathcal{L}_{\text{clf}}(x, y; \theta) = -\log\left(\texttt{SOFTMAX}\left(f_\theta(x)\right)[y]\right)$$

On the other hand, to maximize the log-likelihood of $p(x)$, we introduce a reparameterization $h_\theta(x) = \log \sum_y \exp(f_\theta(x)[y])$:

$$
\begin{aligned}
\log p_\theta(x) &= \log\left(\sum_y p_\theta(x, y)\right) = \log\left(\sum_y \frac{\exp(f_\theta(x)[y])}{\int_{x'} \sum_{y'} \exp(f_\theta(x')[y']) \, dx'}\right) \\
&= \log\left(\frac{\exp(\log(\sum_y \exp(f_\theta(x)[y])))}{\int_{x'} \exp(\log(\sum_{y'} \exp(f_\theta(x')[y']))) \, dx'}\right) = \log\left(\frac{\exp(h_\theta(x))}{\int_{x'} \exp(h_\theta(x')) \, dx'}\right) \\
&= h_\theta(x) - \log(Z'(\theta)),
\end{aligned}
\tag{7}
$$

where $Z'(\theta) = \int_x \exp(h_\theta(x)) \, dx$. Similar to Eq. (5), we can rewrite $\log Z'(\theta)$ by its Fenchel duality:

$$\log Z'(\theta) = \max_q \left[\mathop{\mathbb{E}}_{q(x)}\left[h_\theta(x)\right] + H(q)\right] \tag{8}$$

where $q$ is a distribution of $x$ and $H(q)$ is the entropy of $q$.

Combining Eq. (7) and Eq. (8) and reusing the generator in Section 2.2, we obtain the optimization problem:

$$\max_\theta \min_\phi \mathop{\mathbb{E}}_{p_d(x,y)}\left[h_\theta(x)\right] - \mathop{\mathbb{E}}_{p(z)}\left[h_\theta(q_\phi(z, y))\right] - H(q_\phi) \tag{9}$$

Similar to Eq. (6), the objective of the unconditional discriminator is equivalent to typical GANs augmented with an entropy term. The loss function without considering the entropy can be formulated as:

$$
\begin{aligned}
\mathcal{L}_{d_2}(x, z, y; \theta) &= -h_\theta(x) + h_\theta(q_\phi(z)) \\
\mathcal{L}_{g_2}(z, y; \phi) &= -h_\theta(q_\phi(z, y))
\end{aligned}
$$

## 2.4 Entropy Approximation in cGANs

In Section 2.2 and Section 2.3, we propose two approaches to train cGANs with and without classification. Unsolved problems in Eq. (6) and Eq. (9) are the entropy terms $H(q_\phi(\cdot, y))$ and $H(q_\phi)$. In previous work, various estimators have been proposed to estimate entropy or its gradient [46, 42, 21, 27]. One can freely choose any approach to estimate the entropy in the proposed framework. In this work, we consider two entropy estimators, and we will show how they connect with existing cGANs.

The first approach is the naive constant approximation. Since entropy is always non-negative, we naturally have the constant zero as a lower bound. Therefore, we can maximize the objective by replacing the entropy term with its lower bound, which is zero in this case. This approach is simple but we will show its effectiveness in Section 4 and how it links our framework to ProjGAN and ContraGAN in Section 3.

The second approach is estimating a variational lower bound. Informally, given a batch of data $\{(x_1, y_1), \ldots, (x_m, y_m)\}$, an encoder function $l$, and a class embedding function $e(y)$, the negative 2C loss used in ContraGAN [16],

$$\mathcal{L}_C(x_i, y_i; t) = \log\left(\frac{d(l(x_i), e(y_i)) + \sum_{k=1}^m [\![y_k = y_i]\!] \, d(l(x_i), l(x_k))}{d(l(x_i), e(y_i)) + \sum_{k=1}^m [\![k \neq i]\!] \, d(l(x_i), (l(x_k))}\right), \tag{10}$$

is an empirical estimate of a proper lower bound of $H(X)$ [40], where $d(a, b) = \exp(a^\top b / t)$ is a distance function with a temperature $t$. We provide the proof in Appendix B.

The 2C loss heavily relies on the embeddings $l(x)$ and $e(y)$. Although we only need to estimate the entropy of generated data in Eq. (6) and Eq. (9), we still rely on true data to learn the embeddings in

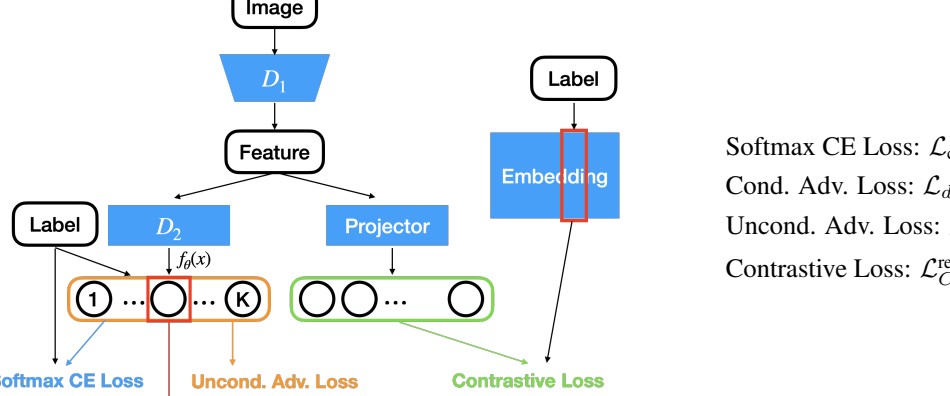

Softmax CE Loss: $\mathcal{L}_{\text{clf}}$
Cond. Adv. Loss: $\mathcal{L}_{d_1}, \mathcal{L}_{g_1}$
Uncond. Adv. Loss: $\mathcal{L}_{d_2}, \mathcal{L}_{g_2}$
Contrastive Loss: $\mathcal{L}_C^{\text{real}}, \mathcal{L}_C^{\text{fake}}$

Figure 1: The discriminator's design of ECGAN. $D_1$ can be any network backbone such as DCGAN, ResGAN, BigGAN. $D_2$ is a linear layer with $K$ outputs, where $K$ is the number of classes.

practice. Therefore, the loss function of Eq. (6) can be written as:

$$\mathcal{L}_{D_1}(x, z, y; \theta) = \mathcal{L}_{d_1}(x, z, y; \theta) + \lambda_c \mathcal{L}_C^{\text{real}}$$
$$\mathcal{L}_{G_1}(z, y; \phi) = \mathcal{L}_{g_1}(x, y; \phi) + \lambda_c \mathcal{L}_C^{\text{fake}},$$

where $\lambda_c$ is a hyperparameter controlling the weight of the contrastive loss, and $\mathcal{L}_C^{\text{real}}, \mathcal{L}_C^{\text{fake}}$ are the contrastive loss calculated on a batch of real data and generated data respectively.

Similarly, the loss function of Eq. (9) becomes:

$$\mathcal{L}_{D_2}(x, z, y; \theta) = \mathcal{L}_{d_2}(x, z, y; \theta) + \lambda_c \mathcal{L}_C^{\text{real}}$$
$$\mathcal{L}_{G_2}(z, y; \phi) = \mathcal{L}_{g_2}(x, y; \phi) + \lambda_c \mathcal{L}_C^{\text{fake}},$$

The introduction of 2C loss allows us to accommodate ContraGAN into our framework.

## 2.5 Energy-based Conditional Generative Adversarial Network

Previous work has shown that multitask training benefits representation learning [30] and training discriminative and generative models jointly outperforms their purely generative or purely discriminative counterparts [11, 28]. Therefore, we propose a framework named Energy-based Conditional Generative Adversarial Network (ECGAN), which combines the two approaches in Section 2.2 and Section 2.3 to learn the joint distribution better. The loss function can be summarized as:

$$\mathcal{L}_D(x, z, y; \theta) = \mathcal{L}_{d_1}(x, z, y; \theta) + \alpha \mathcal{L}_{d_2}(x, z, y; \theta) + \lambda_c \mathcal{L}_C^{\text{real}} + \lambda_{\text{clf}} \mathcal{L}_{\text{clf}}(x, y; \theta) \qquad (11)$$
$$\mathcal{L}_G(z, y; \phi) = \mathcal{L}_{g_1}(z, y; \phi) + \alpha \mathcal{L}_{g_2}(z, y; \phi) + \lambda_c \mathcal{L}_C^{\text{fake}} \qquad (12)$$

where $\alpha$ is a weight parameter for the unconditional GAN loss. The discriminator's design is illustrated in Fig 1.

Here we discuss the intuition of the mechanisms behind each component in Eq. (11). $\mathcal{L}_{d_1}$ is a loss function for conditional discriminator. It updates the $y$-th output when given a data pair $(x, y)$. $\mathcal{L}_{d_2}$ guides to an unconditional discriminator. It updates all outputs according to whether $x$ is real. $\mathcal{L}_{\text{clf}}$ learns a classifier. It increases the $y$-th output and decreases the other outputs for data belonging to class $y$. Finally, $\mathcal{L}_C^{real}$ and $\mathcal{L}_C^{fake}$ play the roles to improve the latent embeddings by pulling the embeddings of data with the same class closer.

Previously, we derive the loss functions $\mathcal{L}_{d_1}$ and $\mathcal{L}_{d_2}$ as the loss in Wasserstein GAN [2]. In practice, we use the hinge loss as proposed in Geometric GAN [26] for better stability and convergence. We use the following combination of $\mathcal{L}_{d_1}$ and $\mathcal{L}_{d_2}$:

$$\text{Hinge}(f_\theta(x_{\text{real}}, y) + \alpha \cdot h_\theta(x_{\text{real}}), f_\theta(x_{\text{fake}}, y) + \alpha \cdot h_\theta(x_{\text{fake}})). \qquad (13)$$

For more discussion of the implementation of hinge loss, please check Appendix C. The overall training procedure of ECGAN is presented in Appendix E.

# 3 Accommodation to Existing cGANs

In this section, we show that our framework covers several representative cGAN algorithms, including ACGAN [39], ProjGAN [35], and ContraGAN [16]. Through the ECGAN framework, we obtain a unified view of cGANs, which allows us to fairly compare and understand the pros and cons of existing cGANs. We name the ECGAN counterparts ECGAN-0, ECGAN-C, and ECGAN-E, corresponding to ProjGAN, ACGAN, and ContraGAN, respectively. We summarize the settings in Table 1 and illustrate the discriminator designs in Appendix F.

| Existing cGAN | ECGAN Counterpart | $\alpha$ | $\lambda_{\text{clf}}$ | $\lambda_c$ |
|---|---|---|---|---|
| ProjGAN | ECGAN-0 | 0 | 0 | 0 |
| ACGAN | ECGAN-C | 0 | $> 0$ | 0 |
| ContraGAN | ECGAN-E | 0 | 0 | $> 0$ |

Table 1: A summary of cGANs and their closest ECGAN counterpart.

## 3.1 ProjGAN

ProjGAN [34] is the most representative cGAN design that is commonly used in state-of-the-art research [3, 50]. Let the output of the penultimate layer in the discriminator be $g(x)$. The output of ProjGAN's discriminator is:

$$D(x, y) = w_u^T g(x) + b_u + w_y^T g(x) = (w_u + w_y)^T g(x) + b_u \tag{14}$$

where $w_u, b_u$ are the parameters for the unconditional linear layer, and $w_y$ is the class embedding of $y$. On the other hand, the output of a discriminator in ECGAN is:

$$D(x, y) = f(x)[y] = (\mathbf{W}^T g(x) + \mathbf{b})[y] = w_y^T g(x) + b_y \tag{15}$$

where $\mathbf{W}, \mathbf{b}$ are the parameters of the linear output layer in $f_\theta$. As shown in Eq. (14) and Eq. (15), the architectures of ProjGAN and ECGAN are almost equivalent. In addition, the loss function of ProjGAN can be formulated as:

$$\mathcal{L}_G = -D(G(z), y)$$
$$\mathcal{L}_D = -D(x, y) + D(G(z), y),$$

which is a special case of ECGAN while $\alpha = \lambda_c = \lambda_{\text{clf}} = 0$. We name this case **ECGAN-0**, which is the simplest version of ECGAN. Compared with ProjGAN, ECGAN-0 has additional bias terms for the output of each class.

## 3.2 ACGAN

ACGAN [39] is the most well-known cGAN algorithm that leverages a classifier to achieve conditional generation. Given a $K$-class dataset, the discriminator of ACGAN is parameterized by a network with $K + 1$ outputs. The first output, denoted as $D(x)$, is an unconditional discriminator distinguishing between real and fake images. The remaining $K$ outputs, denoted as $C(x)$, is a classifier that predicts logits for every class. The loss function of ACGAN can be formulated as:

$$\mathcal{L}_G = -D(G(z)) + \lambda_g \mathcal{L}_{\text{clf}}(G(z), y; C)$$
$$\mathcal{L}_D = -D(x) + D(G(z)) + \lambda_d(\mathcal{L}_{\text{clf}}(x, y; C) + \mathcal{L}_{\text{clf}}(G(z), y; C))$$

where $G$ is the generator, $\lambda_g$ and $\lambda_d$ are hyperparameters to control the weight of cross-entropy loss.

The formulation of ACGAN is similar to our ECGAN when $\alpha = \lambda_c = 0$ and $\lambda_{\text{clf}} > 0$. We call the special case as **ECGAN-C**, with a suffix 'C' for classification loss. ECGAN-C uses a conditional discriminator which plays the role of a classifier at the same time. Hence the generator in ECGAN-C learns from the conditional discriminator rather than the cross-entropy loss which is biased for generative objectives.

| Dataset | # training | # test | # classes | Resolution | # training data per class |
|---|---|---|---|---|---|
| CIFAR-10 | 50,000 | 10,000 | 10 | $32 \times 32$ | 5,000 |
| Tiny ImageNet | 100,000 | 10,000 | 200 | $64 \times 64$ | 500 |
| ImageNet | 1,281,167 | 50,000 | 1,000 | $128 \times 128$ | 1,281 |

Table 2: Datasets for evaluation.

## 3.3 ContraGAN

ContraGAN [16] proposed 2C loss, which we mentioned in Eq. (10), to capture the data-to-data relationship and data-to-label relationship. The 2C loss is applied in both discriminator and generator to achieve conditional generation. That is:

$$\mathcal{L}_G = -D(G(z), y) + \lambda_c \mathcal{L}_C^{\text{fake}}$$
$$\mathcal{L}_D = -D(x, y) + D(G(z), y) + \lambda_c \mathcal{L}_C^{\text{real}}$$

The loss functions are similar to ones in ECGAN with $\alpha = \lambda_{\text{clf}} = 0$ and $\lambda_c > 0$. We call it **ECGAN-E**, where 'E' means entropy estimation. The main difference between ContraGAN and ECGAN-E is the output layer of their discriminators. While ContraGAN uses a single-output network, ECGAN uses a $K$-output network $f_\theta$ which has higher capacity.

We keep Eq. (11) and Eq. (12) as simple as possible to reduce the burden of hyperparameter tuning. Under the simple equations of the current framework, ECGAN-C and ECGAN-E are the closest counterparts to ACGAN and ContraGAN. The subtle difference (in addition to the underlying network architecture) is that ACGAN uses $\mathcal{L}_{d_2}$ instead of $\mathcal{L}_{d_1}$ (ECGAN-C); ContraGAN uses $\mathcal{L}_{d_2}, \mathcal{L}_{g_2}$ instead of $\mathcal{L}_{d_1}, \mathcal{L}_{g_1}$ (ECGAN-E). One future direction is to introduce more hyperparameters in Eq. (11) and Eq. (12) to get closer counterparts.

## 4  Experiment

We conduct our experiments on CIFAR-10 [20] and Tiny ImageNet [22] for analysis, and ImageNet [6] for large-scale empirical study. Table 2 shows the statistics of the datasets. All datasets are publicly available for research use. They were not constructed for human-related study. We do not specifically take any personal information from the datasets in our experiments.

In our experiment, we use two common metrics, Frechet Inception Distance [FID; 14] and Inception Score [IS; 44], to evaluate our generation quality and diversity. Besides, we use **Intra-FID**, which is the average of FID for each class, to evaluate the performance of conditional generation.

### 4.1  Experimental Setup

We use StudioGAN[1] [16] to conduct our experiments. StudioGAN is a PyTorch-based project distributed under the MIT license that provides implementation and benchmark of several popular GAN architectures and techniques. To provide reliable evaluation, we conduct experiments on CIFAR-10 and Tiny ImageNet with 4 different random seeds and report the means and standard deviations for each metric. We evaluate the model with the lowest FID for each trial. The default backbone architecture is BigGAN [3]. We fix the learning rate for generators and discriminators to 0.0001 and 0.0004, respectively, and tune $\lambda_{\text{clf}}$ in $\{1, 0.1, 0.05, 0.01\}$. We follow the setting $\lambda_c = 1$ in [16] when using 2C loss, and set $\alpha = 1$ when applying unconditional GAN loss. The experiments take 1-2 days on single GPU (Nvidia Tesla V100) machines for CIFAR-10, Tiny ImageNet, and take 6 days on 8-GPU machines for ImageNet. More details are described in Appendix D.

### 4.2  Ablation Study

We start our empirical studies by investigating the effectiveness of each component in ECGAN. We use symbols 'U' to represent unconditional GAN loss, 'C' to represent classification loss, and 'E'

---

[1]https://github.com/POSTECH-CVLab/PyTorch-StudioGAN

| Dataset | ECGAN Variant | FID ($\downarrow$) | IS ($\uparrow$) | Intra-FID ($\downarrow$) |
|---|---|---|---|---|
| | ECGAN-0 | $8.049 \pm 0.092$ | $9.759 \pm 0.061$ | $41.708 \pm 0.278$ |
| | ECGAN-U | $\mathbf{7.915} \pm 0.095$ | $9.967 \pm 0.078$ | $41.430 \pm 0.326$ |
| CIFAR-10 | ECGAN-C | $7.996 \pm 0.120$ | $9.870 \pm 0.157$ | $41.715 \pm 0.307$ |
| | ECGAN-UC | $7.942 \pm 0.041$ | $\mathbf{10.002} \pm 0.120$ | $41.425 \pm 0.221$ |
| | ECGAN-UCE | $8.039 \pm 0.161$ | $9.898 \pm 0.064$ | $\mathbf{41.371} \pm 0.278$ |
| | ECGAN-0 | $24.077 \pm 1.660$ | $16.173 \pm 0.671$ | $214.811 \pm 3.627$ |
| | ECGAN-U | $20.876 \pm 1.651$ | $15.318 \pm 1.148$ | $215.117 \pm 7.034$ |
| Tiny ImageNet | ECGAN-C | $24.853 \pm 3.902$ | $16.554 \pm 1.500$ | $212.661 \pm 8.135$ |
| | ECGAN-UC | $\mathbf{18.919} \pm 0.774$ | $\mathbf{18.442} \pm 1.036$ | $\mathbf{203.373} \pm 5.101$ |
| | ECGAN-UCE | $24.728 \pm 0.974$ | $17.935 \pm 0.619$ | $209.547 \pm 1.968$ |

Table 3: Ablation study of ECGAN on CIFAR-10 and Tiny ImageNet. ECGAN-0 means the vanilla version of ECGAN where $\alpha = \lambda_{\text{clf}} = \lambda_c = 0$. The label U stands for unconditional gan loss ($\alpha > 0$). C means classification loss ($\lambda_{\text{clf}} > 0$). E means entropy estimation loss via contrastive learning ($\lambda_c > 0$).

to represent entropy estimation loss, which is 2C loss in our implementation. The concatenation of the symbols indicates the combination of losses. For example, ECGAN-UC means ECGAN with both unconditional GAN loss and classification loss ($\alpha > 0$ and $\lambda_{\text{clf}} > 0$). Table 3 shows the results of ECGAN from the simplest ECGAN-0 to the most complicated ECGAN-UCE. On CIFAR-10, ECGAN-0 already achieves decent results. Adding unconditional loss, classification loss, or contrastive loss provides slightly better or on-par performance. On the harder Tiny Imagenet, the benefit of unconditional loss and classification loss becomes more significant. While ECGAN-U already shows advantages to ECGAN-0, adding classification loss to ECGAN-U further improves all metrics considerably. We also observe that directly adding classification loss is not sufficient to improve cGAN, which is consistent to the finding in [34]. The fact reveals that the unconditional GAN loss is a crucial component to bridge classifiers and discriminators in cGANs. We also find that adding contrastive loss does not improve ECGAN-UC. An explanation is that the entropy estimation lower bound provided by the contrastive loss is too loose to benefit the training. Furthermore, the additional parameters introduced by 2C loss make the optimization problem more complicated. As a result, we use the combination ECGAN-UC as the default option of ECGAN in the following experiments.

### 4.3 Comparison with Existing cGANs

We compare ECGAN to several representative cGANs including ACGAN [39], ProjGAN [34], and ContraGAN [16], with three representative backbone architectures: DCGAN [41], ResNet [13], and BigGAN [3]. Table 4 compares the results of each combinations of cGAN algorithms and backbone architectures. The results show that ECGAN-UC outperforms other cGANs significantly with all backbone architectures on both CIFAR-10 and Tiny ImageNet. We also noticed that ContraGAN, though achieves decent image quality and diversity, learns a conditional generator that interchanges some classes while generating, hence has low Intra-FID. Overall, the experiment indicates that ECGAN-UC can be a preferred choice for cGAN in general situations.

### 4.4 Comparisons between Existing cGANs and their ECGAN Counterpart

Table 5 compares ProjGAN, ContraGAN, ACGAN to their ECGAN counterparts. As we described in Section 3, each of these representative cGANs can be viewed as special cases under our ECGAN framework. As mentioned in Section 3, ECGAN-0 has additional bias terms in the output layer compared to ProjGAN. The results in Table 5 shows that the subtle difference still brings significant improvement to the generation quality, especially on the harder Tiny ImageNet.

Compared to ContraGAN, ECGAN-E has the same loss but different design in the discriminator's output layer. While the discriminator of ContraGAN has only single output, ECGAN-E has multiple outputs for every class. The difference makes ECGAN-E solve the label mismatching problem of ContraGAN mentioned in Section 4.3 and benefits generation on CIFAR-10, but does not work well on Tiny ImageNet. It is probably because of the scarcity of training data in each class in Tiny ImageNet. Only 50 data are available for updating the parameters corresponding to each class.

| Dataset | Backbone | method | FID ($\downarrow$) | IS ($\uparrow$) | Intra-FID ($\downarrow$) |
|---|---|---|---|---|---|
| CIFAR-10 | DCGAN | ACGAN | $32.507 \pm 2.174$ | $7.621 \pm 0.088$ | $129.603 \pm 1.212$ |
| | | ProjGAN | $21.918 \pm 1.580$ | $8.095 \pm 0.185$ | $68.164 \pm 2.055$ |
| | | ContraGAN | $28.310 \pm 1.761$ | $7.637 \pm 0.125$ | $153.730 \pm 9.965$ |
| | | **ECGAN-UC** | $\mathbf{18.035} \pm 0.788$ | $\mathbf{8.487} \pm 0.131$ | $\mathbf{59.343} \pm 1.557$ |
| | ResGAN | ACGAN | $10.073 \pm 0.274$ | $9.512 \pm 0.050$ | $48.464 \pm 0.716$ |
| | | ProjGAN | $10.195 \pm 0.203$ | $9.268 \pm 0.139$ | $46.598 \pm 0.070$ |
| | | ContraGAN | $10.551 \pm 0.976$ | $9.087 \pm 0.228$ | $138.944 \pm 12.582$ |
| | | **ECGAN-UC** | $\mathbf{9.244} \pm 0.062$ | $\mathbf{9.651} \pm 0.098$ | $\mathbf{43.876} \pm 0.384$ |
| | BigGAN | ACGAN | $8.615 \pm 0.146$ | $9.742 \pm 0.041$ | $45.243 \pm 0.129$ |
| | | ProjGAN | $8.145 \pm 0.156$ | $9.840 \pm 0.080$ | $42.110 \pm 0.405$ |
| | | ContraGAN | $8.617 \pm 0.671$ | $9.679 \pm 0.210$ | $114.602 \pm 13.261$ |
| | | **ECGAN-UC** | $\mathbf{7.942} \pm 0.041$ | $\mathbf{10.002} \pm 0.120$ | $\mathbf{41.425} \pm 0.221$ |
| Tiny ImageNet | BigGAN | ACGAN | $29.528 \pm 4.612$ | $12.964 \pm 0.770$ | $315.408 \pm 1.171$ |
| | | ProjGAN | $28.451 \pm 2.242$ | $12.213 \pm 0.624$ | $242.332 \pm 11.447$ |
| | | ContraGAN | $24.915 \pm 1.222$ | $13.445 \pm 0.371$ | $257.657 \pm 3.246$ |
| | | **ECGAN-UC** | $\mathbf{18.780} \pm 1.291$ | $\mathbf{17.475} \pm 1.052$ | $\mathbf{204.830} \pm 5.648$ |

Table 4: Comparison between cGAN variants with different backbone architectures on CIFAR-10 and Tiny ImageNet

| Dataset | method | FID ($\downarrow$) | IS ($\uparrow$) | Intra-FID ($\downarrow$) |
|---|---|---|---|---|
| CIFAR-10 | ProjGAN | $8.145 \pm 0.156$ | $\mathbf{9.840} \pm 0.080$ | $42.110 \pm 0.405$ |
| | ECGAN-0 | $\mathbf{8.049} \pm 0.092$ | $9.759 \pm 0.061$ | $\mathbf{41.708} \pm 0.278$ |
| | ContraGAN | $8.617 \pm 0.671$ | $9.679 \pm 0.210$ | $114.602 \pm 13.261$ |
| | ECGAN-E | $\mathbf{8.038} \pm 0.102$ | $\mathbf{9.876} \pm 0.036$ | $\mathbf{41.155} \pm 0.277$ |
| | ACGAN | $8.615 \pm 0.146$ | $9.742 \pm 0.041$ | $45.243 \pm 0.129$ |
| | ECGAN-C | $\mathbf{8.102} \pm 0.039$ | $\mathbf{9.980} \pm 0.093$ | $\mathbf{41.109} \pm 0.273$ |
| Tiny ImageNet | ProjGAN | $28.451 \pm 2.242$ | $12.213 \pm 0.624$ | $242.332 \pm 11.447$ |
| | ECGAN-0 | $\mathbf{24.077} \pm 1.660$ | $\mathbf{16.173} \pm 0.671$ | $\mathbf{214.811} \pm 3.627$ |
| | ContraGAN | $\mathbf{24.915} \pm 1.222$ | $\mathbf{13.445} \pm 0.371$ | $257.657 \pm 3.246$ |
| | ECGAN-E | $38.270 \pm 1.174$ | $12.576 \pm 0.405$ | $\mathbf{239.184} \pm 2.628$ |
| | ACGAN | $29.528 \pm 4.612$ | $12.964 \pm 0.770$ | $315.408 \pm 1.171$ |
| | ECGAN-C | $\mathbf{24.853} \pm 3.902$ | $\mathbf{16.554} \pm 1.500$ | $\mathbf{212.661} \pm 8.135$ |

Table 5: Compare between representative cGANs and their ECGAN counterparts.

Last, we compare ECGAN-C to ACGAN. Both of them optimize a GAN loss and a classification loss. However, ECGAN-C combines the discriminator and the classifier, so the generator can directly optimize cGAN loss rather than the classification loss. As a result, ECGAN-C demonstrates better performance on both CIFAR-10 and Tiny ImageNet. In sum, the comparisons show that through the unified view provided by ECGAN, we can improve the existing methods with minimal modifications.

### 4.5 Evaluation on ImageNet

We compare our ECGAN-UC and ECGAN-UCE with BigGAN [3] and ContraGAN [16] on ImageNet. We follow all configurations of BigGAN with batch size 256 in StudioGAN. The numbers in Table 6 are reported after 200,000 training steps if not specified. The results show that ECGAN-UCE outperforms other cGANs dramatically. The comparison between ECGAN-UC and ECGAN-UCE indicates that the 2C loss brings more significant improvement in the ECGAN framework than in ContraGAN. The proposed ECGAN-UCE achieves $8.49$ FID and $80.69$ inception score. To the best of our knowledge, this is a state-of-the-art result of GANs with batch size 256 on ImageNet. Selected generated images are shown in Appendix G.

## 5 Related Work

The development of cGANs started from feeding label embeddings to the inputs of GANs or the feature vector at some middle layer [33, 7]. To improve the generation quality, ACGAN [39] proposes to leverage classifiers and successfully generates high-resolution images. The use of classifiers

| Method | FID($\downarrow$) | IS($\uparrow$) |
|---|---|---|
| BigGAN* | 24.68 | 28.63 |
| ContraGAN* | 25.16 | 25.25 |
| ECGAN-UC | 30.05 | 26.47 |
| ECGAN-UCE | 12.16 | 56.33 |
| ECGAN-UCE (400k step) | **8.49** | **80.69** |

Table 6: Evaluation on ImageNet128×128. (*: Reported by StudioGAN.)

in GANs is studied in Triple GAN [24] for semi-supervised learning and Triangle GAN [9] for cross-domain distribution matching. However, Shu [45] and Miyato and Koyama [34] pointed out that the auxiliary classifier in ACGAN misleads the generator to generate images that are easier to be classified. Thus, whether classifiers can help conditional generation still remains questionable.

In this work, we connect cGANs with and without classifiers via an energy model parameterization from the joint probability perspective. [12] use similar ideas but focus on sampling from the trained classifier via Markov Chain Monte Carlo [MCMC; 1]. Our work is also similar to a concurrent work [11], which improves [12] by introducing Fenchel duality to replace computationally-intensive MCMC. They use a variational approach [19] to formulate the objective for tractable entropy estimation. In contrast, we study the GAN perspective and the entropy estimation via contrastive learning. Therefore, the proposed ECGAN can be treated a complements works compared with [12, 11] by studying a GAN perspective. We note that the studied cGAN approaches also result in better generation quality than its variational alternative [11].

Last, [5] study the connection between exponential family and unconditional GANs. Different from [5], we study the conditional GANs with the focus to provide a unified view of common cGANs and an insight into the role of classifiers in cGANs.

## 6   Conclusion

In this work, we present a general framework Energy-based Conditional Generative Networks (ECGAN) to train cGANs with classifiers. With the framework, we can explain representative cGANs, including ACGAN, ProjGAN, and ContraGAN, in a unified view. The experiments demonstrate that ECGAN outperforms state-of-the-art cGANs on benchmark datasets, especially on the most challenging ImageNet. Further investigation can be conducted to find a better entropy approximation or improve cGANs by advanced techniques for classifiers. We hope this work can pave the way to more advanced cGAN algorithms in the future.

## 7   Limitations and Potential Negative Impacts

There are two main limitations in the current study. One is the investigation on ImageNet. Ideally, more experiments and analysis on ImageNet can further strengthen the contribution. But training with such a large dataset is barely affordable for our computational resource, and we can only resort to the conclusive findings in the current results. The other limitation is whether the metrics such as FID truly reflect generation quality, but this limitation is considered an open problem to the community anyway.

As with any work on generative models, there is a potential risk of the proposed model being misused to create malicious content, much like how misused technology can be used to forge bills. In this sense, more anti-forgery methods will be needed to mitigate the misuse in the future.

## Acknowledgement

We thank the anonymous reviewers for valuable suggestions. This work is partially supported by the Ministry of Science and Technology of Taiwan via the grants MOST 107-2628-E-002-008-MY3 and 110-2628-E-002-013. We also thank the National Center for High-performance Computing (NCHC) of National Applied Research Laboratories (NARLabs) in Taiwan for providing computational resources.

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
