# A   Derivation of Variational Log-Partition Function

$$\max_q \mathbb{E}_{q(x)} \left[ f_\theta(x) \right] + H(q)$$

$$= \max_q \int_x q(x) f_\theta(x) \, dx - \int_x q(x) \log\left( q(x) \right) \, dx$$

$$= \max_q \int_x q(x) \log\left( \frac{\exp\left( f_\theta(x) \right)}{q(x)} \right) \, dx$$

$$= \max_q \int_x q(x) \log\left( \frac{\exp\left( f_\theta(x) \right)}{q(x)} \right) \, dx - \log Z(\theta) + \log Z(\theta)$$

$$= \max_q \int_x q(x) \log\left( \frac{\exp\left( f_\theta(x) \right)/Z(\theta)}{q(x)} \right) \, dx + \log Z(\theta)$$

$$= \max_q -\text{KL}\left( q(x) \| p_\theta(x) \right) + \log Z(\theta)$$

$$= \log Z(\theta)$$

# B   2C Loss as a Variational Lower Bound of Entropy

In Section 2.4 we use 2C loss as a lower bound of the entropy. Here we provide the proof.

Given samples $(x_1, y)$ from $p(x_1)p(y|x_1)$ and additional $M-1$ samples $x_2, \ldots x_M$, Eq. (10) in [40] have shown that the InfoNCE loss [47] is a lower bound of mutual information:

$$I(X;Y) \geq \mathbb{E}\left[ \frac{1}{M} \sum_{i=1}^M \log \frac{\exp(f(x_i, y_i))}{\frac{1}{M} \sum_{j=1}^M \exp(f(x_i, y_j))} \right]$$

where the expectation is over $M$ independent samples from the joint distribution: $\Pi_j p(x_j, y_j)$ and $f$ can be any function.

Let

$$f(x_i, y_j) = \begin{cases} l(x_i)^\top e(y_i)/t, & \text{for } i = j \\ l(x_i)^\top l(x_j)/t, & \text{for } i \neq j, \end{cases}$$

We have

$$I(X;Y) \geq \mathbb{E}\left[ \frac{1}{M} \sum_{i=1}^M \log\left( \frac{\exp\left( l(x_i)^\top e(y_i)/t \right)}{\exp(l(x_i)^\top e(y_i)/t) + \sum_{j=1}^M [\![ i \neq j ]\!] \exp\left( l(x_i)^\top l(x_j)/t \right)} \right) \right],$$

which is Eq. (7) in [16].

Since $H(X) = I(X;Y) + H(X|Y)$ and $H(X|Y) \geq 0$, $H(X) \geq I(X;Y)$. Therefore, 2C loss is a variational lower bound of $H(X)$.

# C   Implementation Issue of Hinge Loss

In Section 2.2 and Section 2.3, we derive the loss functions $\mathcal{L}_{d_1}$ and $\mathcal{L}_{d_2}$ as the loss in Wasserstein GAN [2]. In practice, we use the hinge loss as proposed in Geometric GAN [26] for better convergence. An intuitive combination of $\mathcal{L}_{d_1}$ and $\mathcal{L}_{d_2}$ can be as following:

$$\text{Hinge}(f_\theta(x_{\text{real}}, y), f_\theta(x_{\text{fake}}, y)) + \alpha \cdot \text{Hinge}(h_\theta(x_{\text{real}}), h_\theta(x_{\text{fake}})), \tag{16}$$

where $\text{Hinge}(\cdot)$ is the hinge loss function proposed in [26].

The property of the hinge loss encourages the output value of $f_\theta(x_{\text{real}}, y), h_\theta(x_{\text{real}})$ to 1, and $f_\theta(x_{\text{fake}}, y), h_\theta(x_{\text{fake}})$ to $-1$, which leads to better stability in optimization generally. However, since $h_\theta(x) = \log \sum_y \exp(f_\theta(x)[y])$, we notice that encouraging the output of both $f_\theta, h_\theta$ into the same scale harms the optimization. Therefore, we use the following combination instead:

$$\text{Hinge}(f_\theta(x_{\text{real}}, y) + \alpha \cdot h_\theta(x_{\text{real}}), f_\theta(x_{\text{fake}}, y) + \alpha \cdot h_\theta(x_{\text{fake}})). \tag{17}$$

The new formulation leads to more stable optimization and is less sensitive to the parameter $\alpha$ empirically.

## D   Experimental Setup Details

We use hinge loss [26] and apply spectral norm [35] on all models to stabilize the training. We adopt the self-attention technique [50] and horizontal random flipping [52] to provide better generation quality. We apply moving average update [17, 31, 49] for generators after 1,000 generator updates for CIFAR-10 and 20,000 generator updates for Tiny ImageNet with a decay rate of 0.9999. We follow the setting of 2C-loss in [16], using $\lambda_c = 1$ and 512-dimension linear projection layer for CIFAR-10 and 768-dimension linear projection layer for Tiny ImageNet. We use Adam [19] optimizer with batch size 64 for CIFAR-10 and batch size 256 for Tiny ImageNet. The training takes 150,000 steps for CIFAR-10 and 100,000 steps for Tiny ImageNet.

## E   Training Algorithm

**Input:** Unconditional GAN loss weight: $\alpha$. 2C loss weight: $\lambda_c$. Classification loss weight: $\lambda_{\text{clf}}$. Parameters of the discriminator and the generator: $(\theta, \phi)$.
**Output:** $(\theta, \phi)$

Initialize $(\theta, \phi)$
**for** $\{1, \ldots, n_{iter}\}$ **do**
   **for** $\{1, \ldots, n_{dis}\}$ **do**
      Sample $\{(x_i, y_i)\}_{i=1}^{m} \sim p_d(x, y)$
      Sample $\{z_i\}_{i=1}^{m} \sim p(z)$
      Calculate $\mathcal{L}_D$ by Eq. (11)
      $\theta \longleftarrow \text{Adam}(\mathcal{L}_D, lr_d, \beta_1, \beta_2)$
   **end for**
   Sample $\{(y_i)\}_{i=1}^{m} \sim p_d(y)$ and $\{z_i\}_{i=1}^{m} \sim p(z)$
   Calculate $\mathcal{L}_G$ by Eq. (12)
   $\phi \longleftarrow \text{Adam}(\mathcal{L}_G, lr_g, \beta_1, \beta_2)$
**end for**

## F   Discriminator Designs of Existing cGANs and their ECGAN Counterparts

Fig. 2 depicts the discriminator designs of existing cGANs and their ECGAN counterparts.

## G   Images Generated by ECGAN

Fig. 3, Fig. 4, Fig. 5 shows the images generated by ECGAN for CIFAR-10, Tiny ImageNet, and ImageNet respectively.

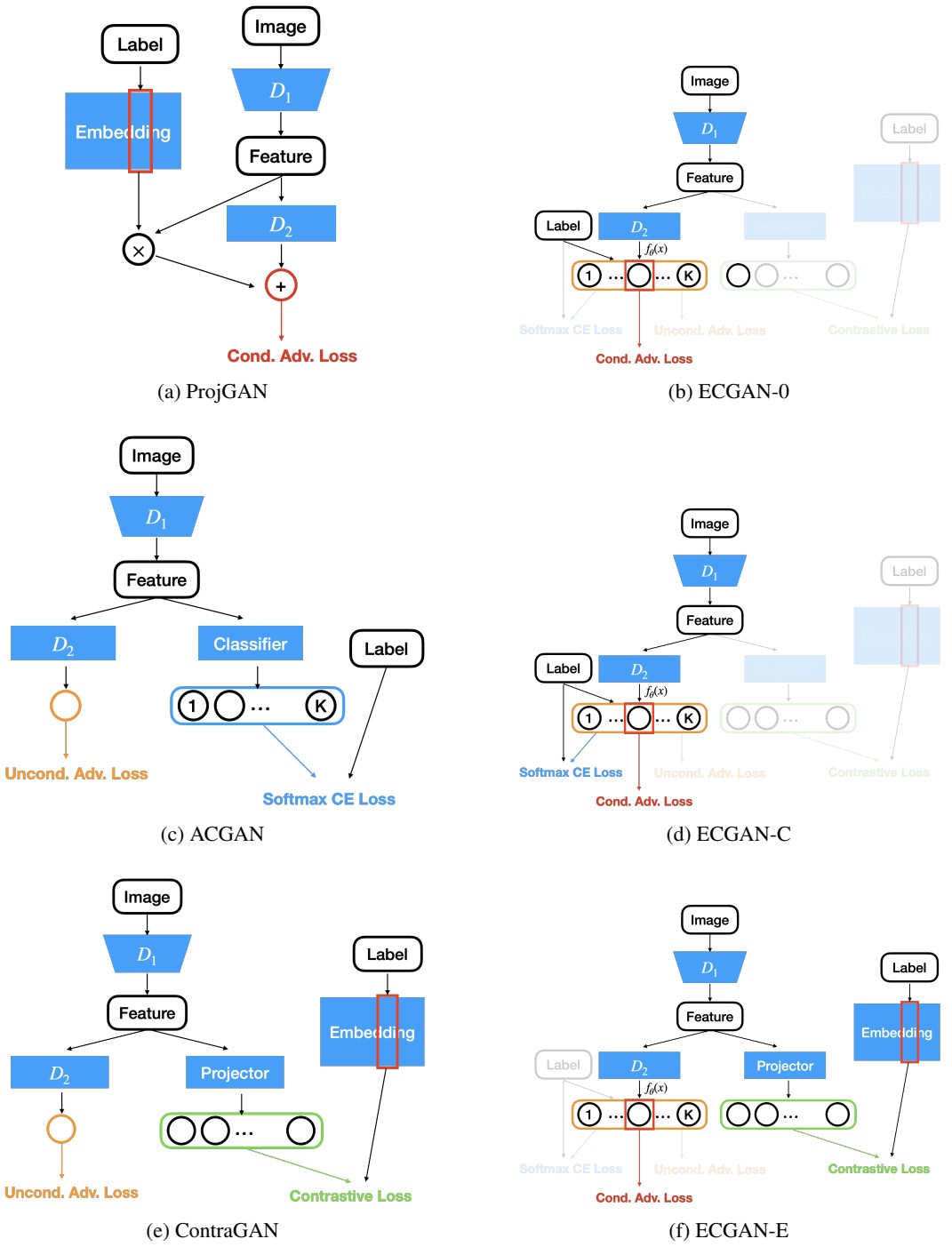

Figure 2: Discriminator Designs of Existing cGANs and their ECGAN Counterparts

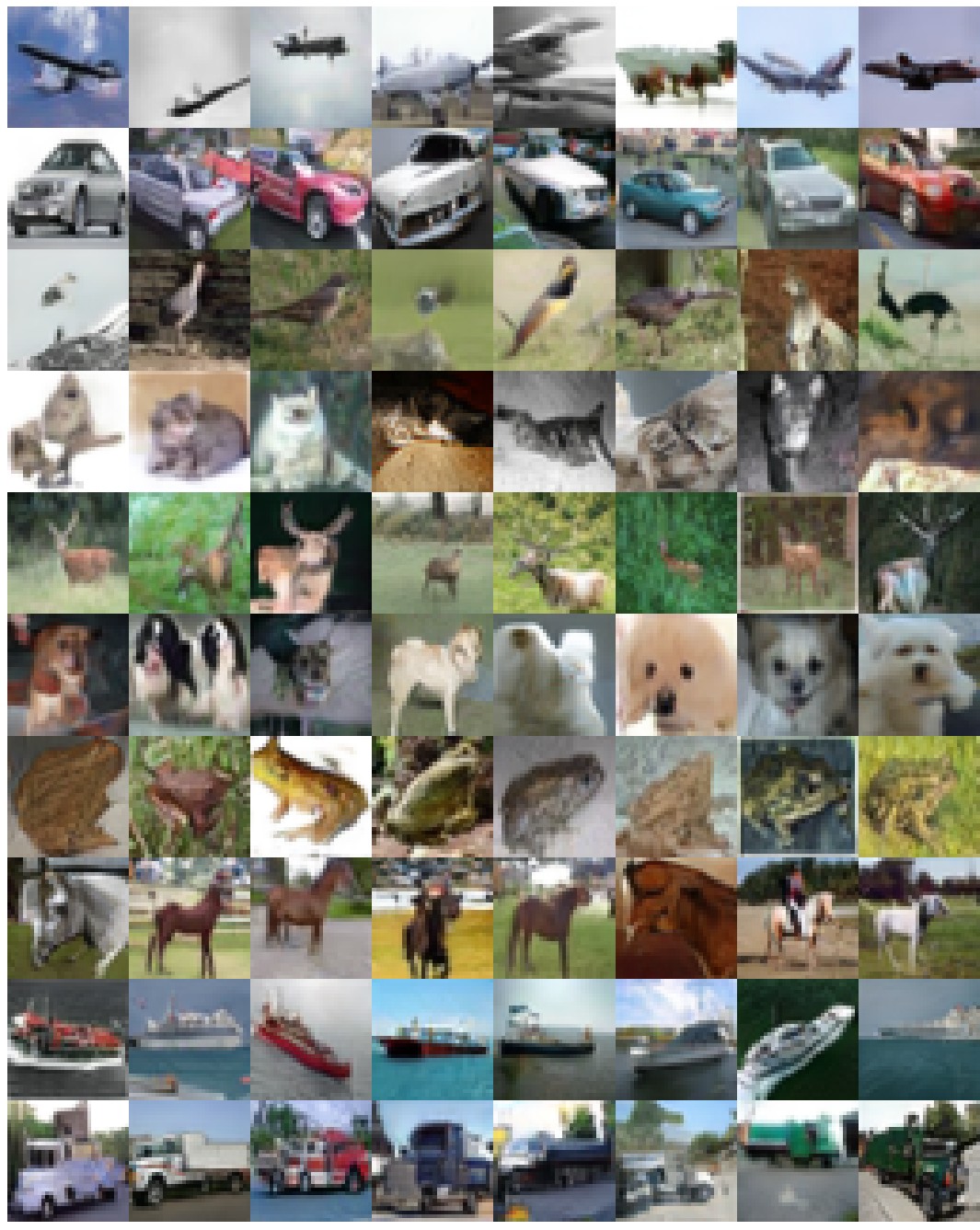

Figure 3: CIFAR-10 images generated by ECGAN-UC (FID: 7.89, Inception Score: 10.06, Intra-FID: 41.42)

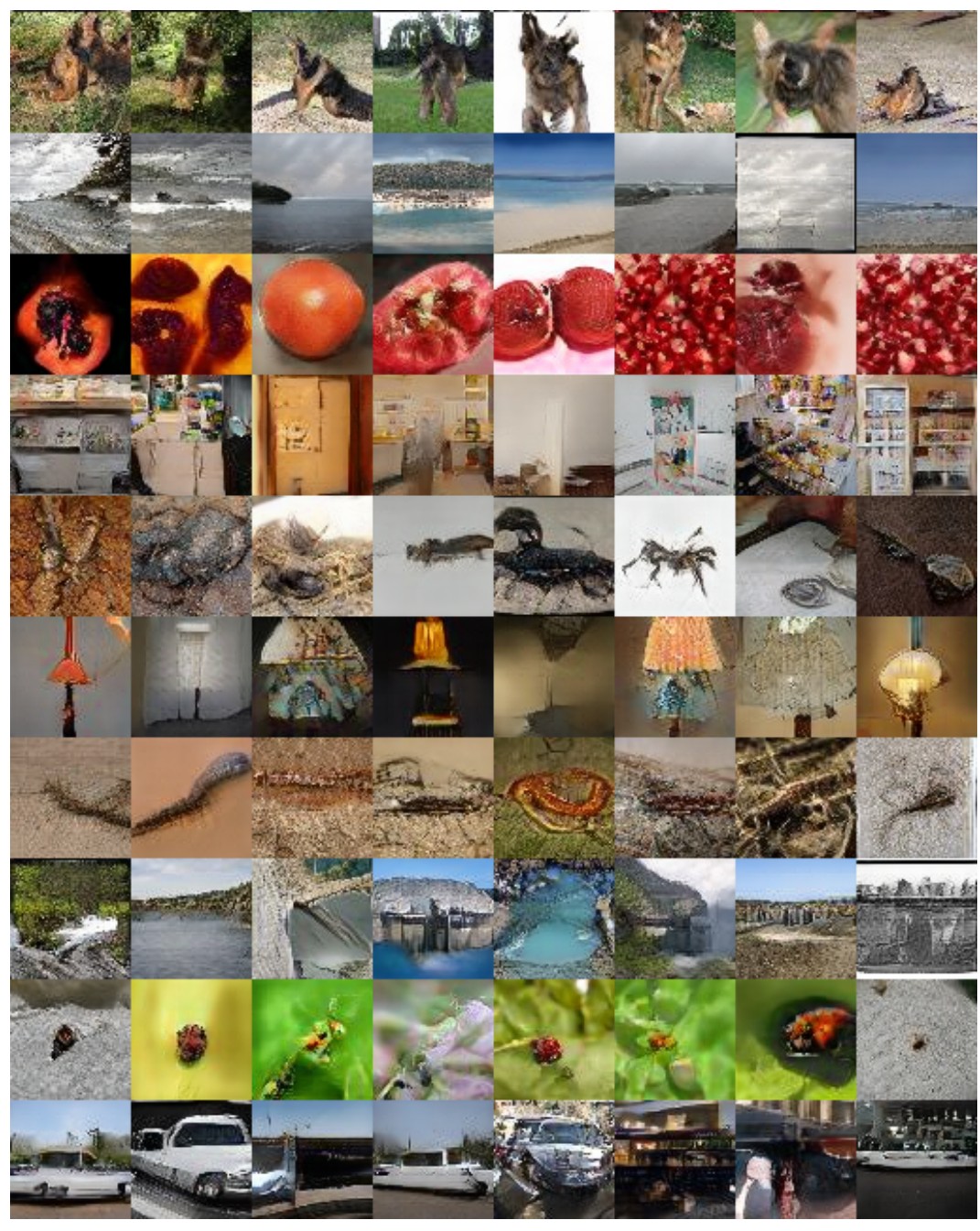

Figure 4: Tiny ImageNet images generated by ECGAN-UC (FID: 17.16, Inception Score: 17.77, Intra-FID: 201.66)

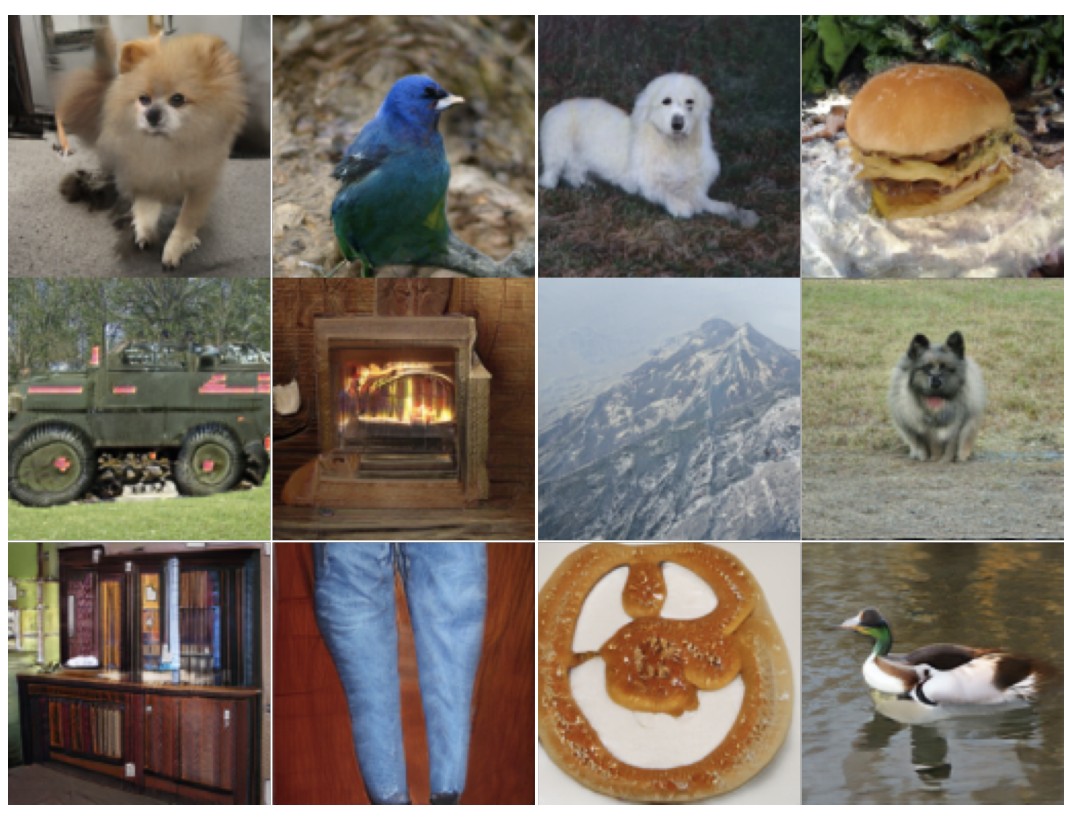

Figure 5: ImageNet images generated by ECGAN-UCE (FID: 8.491, Inception Score: 80.685)