# OpenReview forum: "A Unified View of cGANs with and without Classifiers"
_NeurIPS.cc/2021/Conference — NeurIPS 2021 Poster_

### Official Review · Reviewer_hEZ7 · 2021-07-14

**Rating:** 6
**Confidence:** 3

**Summary:**

The authors discuss that existing methods on including classifiers in a cGAN biases the generator in generating easy to classify images. Therefore, they propose a way to include classifiers in a cGAN to improve its performance in a principled manner. To do so, they decompose the joint probability distribution by the Bayes rule that results in linking classifiers to conditional discriminators. The proposed formulation shows that a joint generator model can be trained from two directions: a conditional discriminator and an un-conditional discriminator with a classifier.
They combine the formulation of these two routes and propose a new method called Energy-based cGAN (ECGAN). ECGAN shows how to use a classifier for cGANs, and it explains other variants of cGANs such as ContraGAN, ACGAN, and ProjGAN. They empirically show that ECGAN outperforms existing cGANs by achieving higher FID (and similar ones) score on two sets of datasets (CIFAR10, Tiny ImageNet).


**Ethical Concerns:**

No ethical concerns

**Limitations And Societal Impact:**

Limitations and societal impact are adequately discussed in the paper.

**Main Review:**

Positives

•	The paper's motivation and the overall idea are interesting.

•	Principled way to use classifiers in conditional GAN. The authors present a new view to explore cGANs and use classifiers in a way that is beneficial to cGANs.

•	The presented framework covers and explains other variants of cGANs (ContraGAN, ProjGAN, ACGAN).

•	Experimental results confirm previous results such as adding a classifier directly to ProjGAN does not result in improvements.

•	Based on the experiments, the ECGAN method achieves higher scores compared to the existing methods.

•	The experimental findings are consistent across different GAN architectures.

Negatives:

•	The additional of the cross entropy loss to equation (11) appears adhoc to cover ACGAN as one of the variants of ECGAN.

•	The authors may add more explanation on the general architecture of the method. Are there two Discriminators with a
 classifier and a generator? There is some confusion regarding whether the classifier and the conditional discriminator are the same neural network or separate?

•	The authors explain how they link conditional discriminators and classifiers. It is, however, unclear why the loss components are helping? what kinds of behaviors are promoted by the loss components?

•	It would be beneficial if the authors also present training stability analysis (similar to figure 3 of ContraGAN [1]). This analysis can reveal how the formulation and losses are helpful.

•	It would be great if the authors can visualize and compare the diversity of generated images across different existing cGANs (ContraGAN, ProjGAN, ACCGAN).

•	What is the computational cost of this method compared to the ContraGAN and ProjGAN?

•	Since the results between the various approaches are not significantly different for one of the 2 datasets used (CIFAR-10), the authors could experiment with more challenging datasets (e.g., ImageNet) to demonstrate that their findings are consistent.


[1] Kang, Minguk, and Jaesik Park. "Contragan: Contrastive learning for conditional image generation." arXiv preprint arXiv:2006.12681 (2020).



**Time Spent Reviewing:**

5

---

> ### Author Response · Authors · 2021-08-10
> **Response to Reviewer hEZ7**
>
> Thank you for your positive comments and constructive suggestions to our paper. We have tried to answer all of your comments as below:
>
> *1. The additional of the cross entropy loss to equation (11) appears adhoc to cover ACGAN as one of the variants of ECGAN.*
>
> Thank you for the comment. We hope to clarify that the cross-entropy loss is not included to explain ACGAN in an ad-hoc manner. We derive the cross-entropy loss based on our idea to learn the joint distribution by connecting the energy function to a classifier, as shown in Section 2.3.
>
> *2. The authors may add more explanation on the general architecture of the method. Are there two Discriminators with a classifier and a generator? There is some confusion regarding whether the classifier and the conditional discriminator are the same neural network or separate?*
>
> Thank you for pointing out the issue. We agree the architecture could be more precise with an illustration. In ECGAN, there is only one discriminator, and it also plays the role of a classifier. To be more specific, we use the neural network $f_\theta$ as stated in line 96 to approximate the energy function, then derive the classifier and both conditional and unconditional discriminators from the energy function $f_\theta$ as in line 113, Eq. (4) and Eq (7). We will clarify those in the revision with a figure and/or more illustrations.
>
> *3. The authors explain how they link conditional discriminators and classifiers. It is, however, unclear why the loss components are helping? what kinds of behaviors are promoted by the loss components?*
>
> About the benefit of combining the loss, our intuition is that optimizing a neural network in two different ways with the same objective would make the network more robust and help the convergence. In our case, the objective is to learn the energy function of the joint distribution, and we derive two different approaches to learn it. Therefore, the more robust energy function brings better joint distributions and conditional generators. While our evaluations focus on Inception score and FID, we did not observe significant behavior changes except for the improvement on both metrics.
>
> *4. It would be beneficial if the authors also present training stability analysis (similar to figure 3 of ContraGAN). This analysis can reveal how the formulation and losses are helpful.*
>
> Thank you for the suggestion. We agree that a stability analysis would help understand the proposed approach. We have conducted such analysis on the side and the findings are similar to those in ContraGAN. We will consider including the findings in the revision if (the precious) space allows.
>
> *5. It would be great if the authors can visualize and compare the diversity of generated images across different existing cGANs (ContraGAN, ProjGAN, ACCGAN).*
>
> Thank you for the suggestion. We agree that visualization may be helpful to understand the difference between the methods. We tried to visualize the results with human eyes, similar to what we presented in Fig. 1 and Fig. 2 of the Appendix. We somehow found it challenging to qualitatively distinguish the generated images from different cGANs with naked human eyes, given that all cGANs have generated images of decent quality. Therefore, we focused more on the quantitative comparisons (Tables 2, 3 and 4) in the paper.
>
> *6. What is the computational cost of this method compared to the ContraGAN and ProjGAN?*
>
> The main overhead of the computation is calculating the contrastive loss. Therefore, the computational cost of ECGAN with contrastive loss (ECGAN-UCE) is similar to that of ContraGAN, and the cost of other variants (ECGAN-0, ECGAN-U, ECGAN-C, ECGAN-UC) is similar to that of ProjGAN.
>
> *7. Since the results between the various approaches are not significantly different for one of the 2 datasets used (CIFAR-10), the authors could experiment with more challenging datasets (e.g., ImageNet) to demonstrate that their findings are consistent.*
>
> Thank you for the suggestion. We agree that relatively small improvement on CIFAR10 has been observed as the performance on CIFAR10 is being saturated. In Table 3, on TinyImageNet, ContraGAN (NeurIPS 2020) is better than ProjGAN (ICLR 2018) by 3.5 FID (28.451->24.915), while the proposed ECGAN improves ContraGAN by 6.1 (24.915->18.780), which we believe to be a non-trivial and non-negligible improvement. We are working on the evaluation on ImageNet and will include the results in the revision. Please check our preliminary results in the common response above.

---

### Official Review · Reviewer_gvcB · 2021-07-16

**Rating:** 6
**Confidence:** 4

**Summary:**

This submission proposes to analyze the most popular variations of conditional GANs (ACGAN, ProjGAN, ContraGAN) under a unified, energy-based, formulation (ECGAN).

Specifically, ECGAN is composed of terms derived from two different decompositions of the joint probability p(x,y).
The paper then links each term to one of the popular cGAN approaches.

ECGAN is evaluated against the traditional models and an extensive ablation study tests the impact of each component of the proposed model.


**Limitations And Societal Impact:**

Limitations and negative impacts are adequately discussed.

**Main Review:**

Originality:
The paper builds very heavily on existing work. The ECGAN formulations are only marginally different from previous works.

Quality:
The submission seems technically sound, supported by adequate theoretical framework and experiments.
It is however difficult to draw conclusive results on only CIFAR-10 and Tiny ImageNet.

Clarity:
The paper is clear and well organized.
The association between the different cGANs and their corresponding ECGAN variants could be debatable, however (see questions below).

Significance:
The proposed formulation provides a unified view of the different losses terms used in conditional GANs that allows for a valuable principled evaluation.
Moreover, considering that ECGAN-UC has very encouraging reported performances and that its implementation is only a slight departure from previous cGAN's, it could become a staple for conditional GANs if future works confirm its good performances.

Overall, I believe the paper is interesting to read and of good quality.
I still have some questions and remarks that I hope can be clarified:

- Since ProjGAN sums both an unconditional and a conditional output, wouldn't it make more sense to compare it to ECGAN-U instead of ECGAN-0? Especially since the used Hinge Loss implementation, described in appendix A, sums the conditional and unconditional terms before the Hinge operation, just like in ProjGAN.
I would appreciate it if the authors could comment and discuss the differences of model and performances between ProjGAN and ECGAN-U.

- Similarly, ACGAN is compared to ECGAN-C in section 3.2. But in section 2.2, ACGAN is already viewed as an unnamed variant of ECGAN (without the ECGAN-0 term and alpha=1 instead).
Could the authors comment on why ACGAN is not directly considered a specific variant of ECGAN?
Considering both previous points, my opinion is that the comparisons in section 3 seem forced and actually make the paper less clear.

- Additional experiments on diverse datasets would strengthen the results, even if performed only with ECGAN-UC and a baseline.


-------------------------------------
Post-rebuttal:
I'd like to thank the authors for their detailled responses and for providing additional results.
After reading the different reviews and responses, I believe the proposed framework is promising and the comparison between the different variants can provide interesting insights for GANs.
However, I also believe the paper needs to be very clear about exactly how the classical cGAN fit or don't fit into the ECGAN formulation.
Without the ability to assess a revised version, I remain unconvinced that small modifications would be sufficient in that regard.

I therefore keep my previous rating (6: Marginally above the acceptance threshold).

**Time Spent Reviewing:**

4.5

---

> ### Author Response · Authors · 2021-08-10
> **Response to Reviewer gvcB**
>
> Thank you for your inspiring questions to our paper. We have tried to answer all of your comments as below:
>
> *1. Since ProjGAN sums both an unconditional and a conditional output, wouldn't it make more sense to compare it to ECGAN-U instead of ECGAN-0? Especially since the used Hinge Loss implementation, described in appendix A, sums the conditional and unconditional terms before the Hinge operation, just like in ProjGAN. I would appreciate it if the authors could comment and discuss the differences of model and performances between ProjGAN and ECGAN-U.*
>
> Thank you for pointing out this issue. As mentioned in Section 3.1, the sum of the unconditional and conditional outputs of ProjGAN (Eq. (14)) is equivalent to the conditional output in ECGAN-0 (Eq. (15)). In contrast, the unconditional output from ECGAN-U is derived from the conditional output with the assumption of energy function (line 117), and does not seem the same as the unconditional output of ProjGAN. Thus, we view ProjGAN to be closer to ECGAN-0 than ECGAN-U.
>
>
> *2. ACGAN is compared to ECGAN-C in section 3.2. But in section 2.2, ACGAN is already viewed as an unnamed variant of ECGAN (without the ECGAN-0 term and alpha=1 instead). Could the authors comment on why ACGAN is not directly considered a specific variant of ECGAN? Considering both previous points, my opinion is that the comparisons in section 3 seem forced and actually make the paper less clear.*
>
> Thank you for pointing out the issue. We cannot directly describe ACGAN as a specific variant of ECGAN because the original network architectures of ACGAN and ECGAN are slightly different. For a $K$-class dataset, ACGAN's discriminator has $K+1$ outputs, with $K$ of them for classification and $1$ for unconditional discrimination. In contrast, ECGAN has only $K$ outputs, each of which plays the role of a conditional discriminator. We will improve our writing by clarifying this subtle difference within Section 3 of the revision.
>
> *3. Additional experiments on diverse datasets would strengthen the results, even if performed only with ECGAN-UC and a baseline.*
>
> Thank you for the suggestion. We are working on the evaluation on ImageNet and will include the results in the revision. Please check our preliminary results in the common response above.

---

### Official Review · Reviewer_YHsT · 2021-07-16

**Rating:** 5
**Confidence:** 3

**Summary:**

This paper introduced a general framework for conditional image generation. And existing cGAN methods such as ACGAN, ProjGAN and ContraGAN are inclued in the proposed algorithm. Experimental results in Cifar and Tiny ImageNet verified the effectiveness of different components of the proposed method.

**Ethics Review Area:**

["Privacy and Security (e.g., consent)"]

**Limitations And Societal Impact:**

1. The experimental results are not convicing. The authors should empoly datasets with higher resolution such as $128\times128$, $256\times256$. The qualitative results in Fig. 1 and Fig. 2 are very blurry.
2. It seems that the reported FID results of existing methods are low compared with the PyTorch-StudioGAN. Please give more explanation.
3. The performance of different variants in Tab. 2 is not consistent. Could the classification loss and entropy estimation loss help the conditional image generation?

**Main Review:**

This workd presented a detailed discussion among the differences on existing condiontal image generation methods. The unified view of cGAN is interesting. However, the authors did not prpose new loss functions for cGANs.


Thanks for the author feedback. Some issues about the experimental evaluation have been addressed welll. However, I still concern about the originality of this work which is also indicated by Reviewer gvcB. In addition, the new results on ImageNet are also confusing and not satisfactory. According to the table below, your reinplementations of ACGAN, ProjGAN and ContraGAN are better. Then the comparisons with BigGAN, ContraGAN based on StudioGAN are not convincing.

**Needs Ethics Review:**

Yes

**Time Spent Reviewing:**

2 hours

---

> ### Author Response · Authors · 2021-08-10
> **Response to Reviewer YHsT**
>
> Thank you for the incisive comments to our paper. We have tried to response to all of your comments as below:
>
> *1. This work presented a detailed discussion among the differences on existing condiontal image generation methods. The unified view of cGAN is interesting. However, the authors did not prpose new loss functions for cGANs.*
>
> Thank you for the comment. Indeed, each component in our loss function originated from existing works, but those existing components were less connected and not fully understood. Our main contribution is to provide a deeper understanding of existing cGANs in a unified manner by decomposing the joint distribution. The unified view allows us to compare the existing cGANs systematically and reveals a principled approach to combine them. By the proper combination, we can achieves better performance than any existing cGAN on the benchmark datasets. We believe that our re-examination and explanation of the existing components can stimulate future research on designing loss functions for cGANs and is valuable to our research community.
>
> *2. The experimental results are not convicing. The authors should empoly datasets with higher resolution such as 128x128, 256x256. The qualitative results in Fig. 1 and Fig. 2 are very blurry.*
>
> Thank you for the suggestion. We are working on the evaluation on ImageNet and will include the results in the revision. Please check our preliminary results in the common response above. Note that the qualitative results in Fig. 1 and Fig. 2 should not be interpreted as the highlights of our work, as it is knowingly difficult by the community to compare GAN-generated images with human eyes. But we have carefully and thoroughly conducted quantative studies that match the standard protocol of the research community [3], and have carefully compared ECGAN with existing cGANs and different ablations, which can be found in Table 2, 3 and 4.
>
>
> *3. It seems that the reported FID results of existing methods are low compared with the PyTorch-StudioGAN. Please give more explanation.*
>
> Thank you for pointing out the issue. The table below shows the Inception Score and FID results of existing methods on TinyImageNet reported by PyTorch-StudioGAN and our paper. The higher IS indicates better performance, and the lower FID indicates better performance, as shown in the table.
>
> The table showcases how we aim for a fair comparison by carefully re-running exisiting cGANs (and getting their best/lowests FID numbers), rather than just quoting the numbers from PyTorch-StudioGAN. During the past years, several techniques were proposed to stabilize or improve the training of GANs, such as spectral normalization [35], Geometric GAN [26], and consistency regularization [47]. The table in PyTorch-StudioGAN follows the original settings of existing works, which means older methods tend to be weaker than newer methods because of not using the new training techniques. The architectures are also different. For example, ACGAN-Mod in the PyTorch-Studio does not use spectral normalization and use ResNet architecture rather than Big-ResNet as in ContraGAN. The difference in these implementation details usually affects the results severly. To make our comparison as fair as possible, we applied those well-established training techniques and architecture to all cGANs. Therefore, our baselines have lower FID scores, which means that they are even harder to beat than the ones in the repository. Please check Appendix B or our code for more implementation details.
>
> | Method        | Reference | IS(⭡)  | FID(⭣) |
> |:------------- |:---------:|:------:|:------:|
> | **ACGAN-Mod** | StudioGAN | 6.342  | 78.513 |
> | **ACGAN**     |   Ours    | 12.964 | 29.528 |
> | **ProjGAN**   | StudioGAN | 6.224  | 89.175 |
> | **ProjGAN**   |   Ours    | 12.213 | 28.451 |
> | **ContraGAN** | StudioGAN | 13.494 | 27.027 |
> | **ContraGAN** |   Ours    | 13.445 | 24.915 |
>
>
> *4. The performance of different variants in Tab. 2 is not consistent. Could the classification loss and entropy estimation loss help the conditional image generation?*
>
> According to our derivation in Section 2, we should optimize the classification and the unconditional loss together to learn the joint distribution. In this case, Table 2 shows consistent results where ECGAN-UC outperforms ECGAN-0 on both datasets. As for the entropy estimation loss, we observe consistent results that adding 2C loss does not improve ECGAN-UC. Please check our response to reviewer iBCw for some possible reasons. We believe that the findings on the 2C loss shed lights on future research that leverages other entropy estimation techniques for cGANs.

---

### Official Review · Reviewer_gBbd · 2021-07-17

**Rating:** 6
**Confidence:** 4

**Summary:**

The authors target to leverage classifiers in a principled manner and unify them in cGAN, by using the decomposition of joint probability distribution and a classic energy model to parameterize the distribution. Experiments show new state-of-the-art generation quality in terms of FID and IS, over several datasets (CIFAR-10, Tiny ImageNet) on several GAN implementations (DCGAN, ResGAN, BigGAN) and against several baselines (ACGAN, ProjGAN, ContraGAN).

**Limitations And Societal Impact:**

Yes

**Main Review:**

********************************************************************
Strengths
********************************************************************
+ Clear exposition: good writing and good result demos.
+ The research problem is clearly defined.
+ Reproducible.
+ Experiments are dense and results are informative to set the new state-of-the-art.

********************************************************************
Weaknesses
********************************************************************

- Experiments can be more convincing if not only on toy datasets like CIFAR-10 or Tiny ImageNet. Repeat the quantitative comparisons on the entire ImageNet dataset with the full resolution.

**Time Spent Reviewing:**

4 hours

---

> ### Author Response · Authors · 2021-08-10
> **Response to Reviewer gBbd**
>
> Thank you for the comments and the positive recognition of our strength. Please check our response to your comments below:
>
> *Experiments can be more convincing if not only on toy datasets like CIFAR-10 or Tiny ImageNet. Repeat the quantitative comparisons on the entire ImageNet dataset with the full resolution.*
>
> Thank you for the suggestion. We are working on the evaluation on ImageNet and will include the results in the revision. Please check our preliminary results in the common response above.

---

### Official Review · Reviewer_iBCw · 2021-08-03

**Rating:** 6
**Confidence:** 3

**Summary:**

The authors of the paper propose a unified view of conditional generative adversarial networks (cGANs). To this end, they analyze the log of joint probability distribution p(x, y) through two different perspective of views; via a conditional discriminator using Eq.(2), and via an unconditional discriminator and classifier using Eq.(3). Inspired by a multitask learning, they suggest Energy-based Conditional Generative Adversarial Networks (ECGAN) where two approaches are combined together to approximate the joint probability distribution. Under this framework, previous cGANs such as ProjGAN, ACGAN and ContraGAN can be closely explained using the variants of ECGAN with the right choice of hyperparameters, namely, ECGAN-0, ECGAN-C and ECGAN-E, respectively.
Their experiments on CIFAR-10 and Tiny ImageNet show that these ECGAN variants outperform ProjGAN, ACGAN and ContraGAN in most cases of experiments. Furthermore, ECGAN-UC outperforms these variants of ECGAN as well as previous cGANs with various backbone structures such as DCGAN, ResGAN, and BigGAN. Through this result, the authors conclude that adding a classifier helps improving cGANs with the help of the unconditional discriminator as opposed to the previous findings in Shu et al. [42] and ProjGAN [34].

**Ethical Concerns:**

The authors acknowledged that the proposed algorithm can be potentially misused through creating malicious contents.

**Limitations And Societal Impact:**

Some limitations that the authors did not recognize are provided above.

**Main Review:**

Originality:

Previous cGANs have taken one of approaches explained in section 2.2 and 2.3 in a broad perspective. By combining them, this paper indeed provides a unified picture of cGANs. Although the concurrent work [11] is somewhat similar in that they both try to solve Eq.(5), the tasks they are trying to solve and the approaches they are taking are quite different as mentioned in the related work.

Quality:
- Line 249 ~ 250 - The authors mentioned that the entropy estimation lower bound provided by the contrastive loss is too loose to benefit the training. But, doesn’t ECGAN-UC correspond to using 0 as a lower bound for entropy (1st approach mentioned to estimate entropy) which is a looser bound for entropy?
- Line 247 ~ 248 - The authors claimed that the unconditional GAN loss is a crucial component to bridge classifiers and discriminators in cGANs according to the performance results provided in Table2. Could authors provide any idea why it is the case other than the empirical evidence?
- Their experiments are somewhat limited as they did not provide the experiment results on ImageNet which were used by all of ProjGAN, ACGAN and ContraGAN.

Clarity:
- Line 101 - It would be better to specify a section of the book or provides a proof in the appendix as done in appendix A.1 of [11].
- Line 141 - Could authors elaborate why Eq.(10) is an empirical estimate of a proper lower bound of an entropy?
- Line 164 ~ 165 - It would be better if the authors can provide experiment results that compare the performance with the original wassertein loss versus hinge loss since it is hard to capture how much improvement is originated from using the hinge loss.
- Line 197 ~ 198 - The authors said the main difference between ACGAN and ECGAN-C is that ECGAN-C uses a conditional discriminator. But, if my understanding on the equations in section 3.2 is correct, another difference is that ACGAN has additional cross entropy loss for the generated images. Do authors think it is negligible? Also, wouldn’t ECGAN-C closer to ACGAN if unconditional discriminator is used instead of conditional discriminator? According to the claim in line 247 ~ 248 that unconditional GAN loss is crucial component to bridge classifiers and discriminators in cGANs, I am not sure why ECGAN-C is set in that way.
- Typos:
    * It seems the denominator part of Eq.(10) is missing an indicator function of (k != I) according to the Eq.(8) in ContraGAN.
    * Line 161 - increase -> increases, decrease -> decreases
    * Eq.(14) - Third term in the middle h(x) -> g(x)
    * Table 4 - bolding is missing for some cells.
    * Citations [11] and [12] are mixed up in the related work section.

Significance:
- Although there are some remaining questions as specified above and experiments are somewhat limited, this work can benefit the community by providing a unified framework for cGANs.

**Time Spent Reviewing:**

12

---

> ### Author Response · Authors · 2021-08-10
> **Response to Reviewer iBCw**
>
> Thank you for your careful reading and suggestions to our paper. We have tried to answer all of your comments as below:
>
> *1. Line 249 ~ 250 - The authors mentioned that the entropy estimation lower bound provided by the contrastive loss is too loose to benefit the training. But, doesn't ECGAN-UC correspond to using 0 as a lower bound for entropy (1st approach mentioned to estimate entropy) which is a looser bound for entropy?*
>
> Yes, we agree that the lower bound provided by ECGAN-UC is looser than the one provided by the constrastive loss. Somehow empirically the tighter bound did not always lead to performance improvement. Our explanation is that the additional network parameters introduced by the 2C loss may make the optimization more challenging. In particular, the 2C loss depends on the learned label embeddings, but the scarcity of data in each class (especially on TinyImageNet) may cause lower-quality embeddings and in term harm the performance when using the 2C loss. Also, accurately estimating entropy through the contrastive formulation requires a large batch size (e.g. thousands in representation learning) to alleviate the bias in addition to the variance [37], while a smaller batch size that follows common experiment protocols and reasonable computation budgets was used in our experiments. The issues above can undermine the benefits of the tigher bound for entropy estimation.
>
> *2. Line 247 ~ 248 - The authors claimed that the unconditional GAN loss is a crucial component to bridge classifiers and discriminators in cGANs according to the performance results provided in Table2. Could authors provide any idea why it is the case other than the empirical evidence?*
>
> At the intution level, aggregated conditional distributions should be equivalent to the uncondition distribution. So adding the unconditional GAN loss does not change the optimum, and the loss can serve as a constraint to respect the equivalence. Such a constraint encourages convergence, as researchers have observed across many other deep learning models.
>
> Mathematically speaking, the unconditional loss is derived from our novel perspective of the energy model under the ECGAN framework. The derivation reveals that the unconditional loss can assist the classifier by preventing it (trained from the cross-entropy loss) to fall into the folklore trap of being overly confident on training examples. This in term leads to a more robust classifier and a more robust discriminator (from the same network).
>
> *3. Their experiments are somewhat limited as they did not provide the experiment results on ImageNet which were used by all of ProjGAN, ACGAN and ContraGAN.*
>
> Thank you for the comment. We are working on the evaluation on ImageNet and will include the results in the revision. Please check our preliminary results in the common response above.
>
> *4. Line 101 - It would be better to specify a section of the book or provides a proof in the appendix as done in appendix A.1 of [1].*
>
> Thank you for the suggestion. We will include the proof in the appendix of the revision.
>
> *5. Line 141 - Could authors elaborate why Eq.(10) is an empirical estimate of a proper lower bound of an entropy?*
>
> According to Eq. 10 in [37], the InfoNCE loss is a lower bound of mutual information between two random variables. By setting the random variables as the generated image distribution and adequately setting the function $f$ in Eq. 10, we can obtain a lower bound of an entropy. We will include this derivation in the revision.
>
> *6. Line 164 ~ 165 - It would be better if the authors can provide experiment results that compare the performance with the original wassertein loss versus hinge loss since it is hard to capture how much improvement is originated from using the hinge loss.*
>
> Thank you for the suggestion. The improvement from Wasserstein loss to hinge loss has been studied in the literature [26]. To make the comparison as fair as possible, we used the hinge loss for all the models in our experiments. That is, all the improvements of ECGAN originate from the modifications proposed in this work.
>
> *7. Line 197 ~ 198 - The authors said the main difference between ACGAN and ECGAN-C is that ECGAN-C uses a conditional discriminator. But, if my understanding on the equations in section 3.2 is correct, another difference is that ACGAN has additional cross entropy loss for the generated images. Do authors think it is negligible?*
>
> Yes, while both ACGAN and ECGAN-C have cross entropy loss for real images, ACGAN has additional cross entropy loss for the generated images. In our side experiments, we have tried adding the cross entropy loss for the generated images to ECGAN-C. The results are slightly worse than not adding the cross entropy loss for generated images but are negligible compared to the difference between ACGAN and ECGAN-C.
>
> *8. Wouldn't ECGAN-C closer to ACGAN if unconditional discriminator is used instead of conditional discriminator? According to the claim in line 247 ~ 248 that unconditional GAN loss is crucial component to bridge classifiers and discriminators in cGANs, I am not sure why ECGAN-C is set in that way.*
>
> We agree that using an unconditional discriminator would make ECGAN-C look closer to ACGAN. We have also tried the setting that $\alpha = 1$ and dropped $\mathcal{L}\_{d_1}, \mathcal{L}\_{g_1}$ in Eq. (11) and Eq. (12). However, we soon discovered that it leads to an unconditional GAN rather than a cGAN because there is no conditional information for the generator. The combination closest to ACGAN would be using asymmetric losses for the generator and the discriminator. That is, using only $\mathcal{L}_\{d_2}$ in Eq. (11) and $\mathcal{L}\_{g_1}$ in Eq. (12). However, the asymmetric combination is less meaningful in our derivation for joint distribution. Therefore, we decided to stay with the setting of Eq. (11) and Eq. (12).
>
> *9. Typos*
>
> Thank you for pointing out the typos. We will fix them in the revision.

---

### Review · Ethics_Reviewer_TQwK · 2021-08-09

**Recommendation:** I don't recommend any changes based o…

**Ethics Review:**

The authors acknowledge that cGANs could be misused with malicious intent so there is some ethical considerations when improving state of the art. This doesn't seem particularly unique to cGANS and in my opinion this type of concern does not preclude acceptance to Neurips. The datasets used are all publicly available and accessible under an MIT license.

---

### Review · Ethics_Reviewer_7bEy · 2021-08-10

**Recommendation:** N/A

---

### Author Response · Authors · 2021-08-10
**Common response and preliminary results on ImageNet**

We thank all reviewers for the positive comments and constructive suggestions on our work. We particularly thank reviewers iBCw and hEZ7 for recognizing the main contributions of our work: providing a deeper understanding of existing cGANs in a unified manner by decomposing the joint distribution. The unified view allows combining existing cGANs in a principled manner and opens a new possibility to achieve better generation performance. Along the line of deeper understanding, we focus on comparing existing cGANs fairly by tuning the hyperparameters of both existing cGANs and our ECGAN carefully to generate extensive and conclusive results in Table 3, which clearly demonstrates the superiority of the proposed ECGAN approach.

Several reviewers share a common suggestion to include evaluation results on the full ImageNet. We certainly appreciate the suggestion and agree that it is important to further understand the competitiveness of ECGAN on more challenging data sets. Somehow it is overly resource consuming at our end to get extensive results like Table 3 on ImageNet (either prior to the submission deadline, or in the reasonable future). As a first step towards incorporating your suggestion, we provide some preliminary results of untuned ECGAN-UCE on ImageNet and discuss our findings below.

We follow the hyperparameter settings of BigGAN256 and ContraGAN in PyTorch-StudioGAN and use the same ECGAN-specific parameters as ones evaluated on TinyImageNet.

| Method                | Reference | IS(⭡)  |
|:--------------------- |:---------:|:------:|
| **BigGAN**            | StudioGAN | 28.633 |
| **ContraGAN**         | StudioGAN | 25.249 |
| **Untuned ECGAN-UCE** |           | 32.187 |

The preliminary results show that the untuned ECGAN-UCE is already competitive to BigGAN and ContraGAN. Note that for ImageNet, the scales of the cross entropy loss and unconditional GAN loss should be different from those on TinyImageNet. That is, there is a room of tuning the weights on those loss terms for ImageNet to achieve even better performance. We plan to keep working on ImageNet and include more results in our revision.

---

> ### Comment · Area_Chair_QKW2 · 2021-09-14
> **FIDs?**
>
> Hello, authors:
>
> Sorry for the very late comment here, but: it seems strange to me that in the paper you included FID and intra-FID scores for all your results, yet this comment only included Inception scores. Can you post the FIDs for this new result, as well?
>
> Thanks!

---

> > ### Author Response · Authors · 2021-09-15
> > **Response to Area Chair**
> >
> > Dear Area Chair,
> >
> > Thank you for asking. The numbers that you request for ECGAN and BigGAN are shown in the table below. They strengthen our conclusion that ECGAN is able to enhance BigGAN, by improving Inception Score from 13.097 to 32.187 and FID from 64.661 to 32.596. Intra-FID on ImageNet is very time-consuming to calculate (requiring > 2 days per number) and is hence not available yet.
> >
> > | Method    | IS(⭡)  | FID(⭣) |
> > | --------- | ------ | ------ |
> > | ECGAN-UCE | 32.187 | 32.596 |
> > | BigGAN | 13.097 | 64.661 |
> > | --------- | ------ | ------ |
> > | BigGAN (reported by StudioGAN) |28.633 |24.684 |
> >
> > After we showed the initial IS results on ECGAN, we spent the whole month carefully tuning both ECGAN and BigGAN on ImageNet to make sure that those numbers are credible. The thing that drew our attention in the past month was that we were unable to reproduce the numbers reported by StudioGAN. The issue has also been raised by several other users of StudioGAN ([issue #40](https://github.com/POSTECH-CVLab/PyTorch-StudioGAN/issues/40), [issue #64](https://github.com/POSTECH-CVLab/PyTorch-StudioGAN/issues/64)), and is likely related to multi-GPU training. We ran our CIFAR-10 and TinyImageNet results (in the paper) on a single GPU, but the size of ImageNet requires multi-GPU training, and the current issue of StudioGAN (and the time-consuming nature) made it significantly harder to reproduce the numbers. We took our best efforts to make sure that the top two rows of the table above are credible numbers under a fair comparison, though.
> >
> > Given that getting a single ImageNet number with our limited computational resources, even when using 8 GPUs, takes a week to run, we believe that it is not fair nor carbon-emission-friendly to require us to reproduce the whole ImageNet result. The numbers above demonstrate that ECGAN holds a clear advantage over BigGAN. This result echoes the observed effectiveness of ECGAN on CIFAR-10 and TinyImageNet, and further confirms the potential of ECGAN. We plan to add those numbers to the revision with a discussion to share our experience on the computational limitation and reproducibility issue with the community.

---

### Decision · Program_Chairs · 2021-09-27

**Decision:**

Accept (Poster)

**Comment:**

This paper proposes a framework for thinking about several existing class-conditional GAN approaches in a unified way, which also leads to a new suggested algorithm. The algorithm outperforms existing approaches on TinyImageNet, and some initial results are shown in the rebuttal that it also does so on ImageNet. I think the framework is sensible and, from what we can tell, the new algorithm is probably an improvement over existing ones. Please ensure that you address the various concerns raised by reviewers, as well as adding the new results, in your final version.

---

> ### Public Comment · ~Si-An_Chen1 · 2021-11-10
> **Updated results on ImageNet**
>
> Dear Revewers and Area Chair,
>
> Thank you for accepting the work. We are happy to let you know that after the discussion period, we discovered and corrected some data preprocessing issues when running on ImageNet. We then succesfully reproduced the result of BigGAN256 on ImageNet with IS 30.65 and FID 22.65, which are close to the numbers reported in StudioGAN (IS 28.63 and FID 24.68).
>
> With the reproducible results of BigGAN256, we move on to update ECGAN's results on ImageNet in our camara-ready revision.
>
> | Method    | IS(⭡) | FID(⭣) |
> | --------- | ----- | ------ |
> | ECGAN-UC  | 26.47 | 30.05  |
> | ECGAN-UCE | 56.33 | 12.16  |
> | ECGAN-UCE (400k step)          |   **80.69**    |   **8.49**     |
>
> As shown in the paper, the numbers further confirm the competiveness of ECGAN. Thank you for valuable suggestions on including ImageNet.